

# Seqenv: linking sequences to environments through text mining

Lucas Sinclair[1,*], Umer Z. Ijaz[2,*], Lars Juhl Jensen[3], Marco J.L. Coolen[4], Cecile Gubry-Rangin[5], Alica Chroňáková[6], Anastasis Oulas[7,8], Christina Pavloudi[8], Julia Schnetzer[9], Aaron Weimann[10], Ali Ijaz[11], Alexander Eiler[1], Christopher Quince[12] and Evangelos Pafilis[8]

[1] Department of Ecology and Genetics, Limnology, Uppsala University, Uppsala, Sweden
[2] Infrastructure and Environment Research Division, School of Engineering, University of Glasgow, Glasgow, United Kingdom
[3] The Novo Nordisk Foundation Center for Protein Research, Faculty of Health and Medical Sciences, University of Copenhagen, Copenhagen, Denmark
[4] Western Australia Organic and Isotope Geochemistry Centre (WA-OIGC), Department of Chemistry, Curtin University of Technology, Bentley, WA, Australia
[5] Institute of Biological & Environmental Sciences, University of Aberdeen, Aberdeen, United Kingdom
[6] Institute of Soil Biology, Biology Centre, Czech Academy of Sciences, České Budějovice, Czech Republic
[7] Bioinformatics Group, The Cyprus Institute of Neurology and Genetics, Nicosia, Cyprus
[8] Institute of Marine Biology Biotechnology and Aquaculture (IMBBC), Hellenic Centre for Marine Research (HCMR), Heraklion Crete, Greece
[9] Department of Molecular Ecology, Microbial Genomics and Bioinformatics Group, Max Planck Institute for Marine Microbiology, Bremen, Germany
[10] Computational Biology of Infection Research, Helmholtz Centre for Infection Research, Braunschweig, Germany
[11] Hawkesbury Institute for the Environment, University of Western Sydney, Hawkesbury, Sydney, Australia
[12] Warwick Medical School, University of Warwick, Warwick, United Kingdom
[*] These authors contributed equally to this work.

Corresponding authors
Christopher Quince,
c.quince@warwick.ac.uk,
christopher.quince@glasgow.ac.uk
Evangelos Pafilis, pafilis@hcmr.gr

## ABSTRACT

Understanding the distribution of taxa and associated traits across different environments is one of the central questions in microbial ecology. High-throughput sequencing (HTS) studies are presently generating huge volumes of data to address this biogeographical topic. However, these studies are often focused on specific environment types or processes leading to the production of individual, unconnected datasets. The large amounts of legacy sequence data with associated metadata that exist can be harnessed to better place the genetic information found in these surveys into a wider environmental context. Here we introduce a software program, seqenv, to carry out precisely such a task. It automatically performs similarity searches of short sequences against the "nt" nucleotide database provided by NCBI and, out of every hit, extracts–if it is available–the textual metadata field. After collecting all the isolation sources from all the search results, we run a text mining algorithm to identify and parse words that are associated with the Environmental Ontology (EnvO) controlled vocabulary. This, in turn, enables us to determine both in which environments individual sequences or taxa have previously been observed and, by weighted summation of those results, to summarize complete samples. We present two demonstrative applications of seqenv to a survey of ammonia oxidizing archaea as well as to a plankton paleome dataset from the Black Sea. These demonstrate the ability of the tool to reveal novel patterns in HTS
and its utility in the fields of environmental source tracking, paleontology, and studies of microbial biogeography. To install seqenv, go to: https://github.com/xapple/seqenv.

## INTRODUCTION

The annotation of DNA sequences, i.e., attaching meaningful labels to them, is key to the interpretation of genomics data. In essence, this process gives context to a sequence. For instance, annotation reveals the taxon from which the sequence was derived (*Wang et al., 2007*) and/or the gene families and potential functions (*Juncker et al., 2009*). However, one type of annotation for which no automated bioinformatics pipeline currently exists is the annotation to the environmental source. In other words, determining the types of environment in which a given sequence has previously been found. We introduce a new program titled "seqenv" which addresses this gap, automatically labeling sequences to the Environmental Ontology (EnvO) (*Buttigieg et al., 2013*). We apply this bioinformatics pipeline to two datasets of environmental marker genes derived from terrestrial archaeal ammonia oxidizers (AOA) (*Gubry-Rangin et al., 2011*) and the Black Sea plankton paleome (*Coolen et al., 2013*). This method reveals hitherto unknown patterns in AOA diversity, and adds to our understanding of the geological history of the Black Sea.

Annotating sequences to environments has become increasingly relevant as a result of the growing application of environmental genomics to microbiology. In environmental genomics, microbial DNA is extracted directly from an environment and then sequenced, possibly following PCR amplification of target marker genes such as the 16S rRNA gene (*Logares et al., 2012*). The result is a catalog of the microorganisms present in a particular sample. One of the first interrogations concerning such samples is to know what other environments these organisms have been found in. The answer can reveal ecologically relevant insight about those organisms and may provide evidence for contamination from other environments. There exists a wealth of information in available databases (most notably the ones provided by NCBI) which can be used to gain a detailed overview of the biogeography of particular sequence varieties. The strategy adopted in seqenv is to take input sequences and match them against the NCBI's database using the time-tested BLAST search algorithm (*Altschul et al., 1990*).

All hits within a level of identity approximating to species are kept and either the text field "isolation source" extracted or the PubMed abstracts associated with the submission obtained. In general, we have found the isolation source metadata to be the most dependable source of environmental information and the results presented here are restricted to that field. A custom named entity recognition (NER) system based on *Pafilis et al. (2015)* is then used to label the resulting text with terms from the EnvO ontology (*Buttigieg et al., 2013*). An ontology is a formal specifications of the terms in a particular knowledge domain and the relations among them. Ontologies are often represented as an acyclic directed graph. The

Environmental Ontology (http://environmentontology.org/) (or EnvO) provides an ontology for this concise, controlled vocabulary for the description of environments. EnvO also has the appeal of having been adopted by the Genomics Standards Consortium for metadata associated with environmental sequence submission (*Field et al., 2014*). The terms found associated with each sequence are then collated together to provide its environmental context.

This environmental annotation scheme can be applied to any type of sequence, protein coding or ribosomal RNA. The sequences can be derived from a particular taxonomic grouping but they can also correspond to operational taxonomic units (OTUs) used as proxies for taxa in environmental sequencing studies (*Blaxter et al., 2005*). In either case, the nature and diversity of environments associated with a particular microorganism can elucidate and bring light to its ecology. Additionally, if OTUs are used, seqenv can also incorporate their abundances across samples. This furnishes a sample-level description of the EnvO terms produced by simply summing the terms associated with each OTU weighted by their relative abundance in the sample. These tables can then be used as a basis for multivariate statistics that contrast communities in terms of the environmental terms associated with their constituent organisms. This novel approach is a powerful means for exploring sample level differences in the origin of community constituents.

Recently, a method has been developed for automatically associating geographic longitude and latitude coordinates to Genbank records through rule based text mining of associated PubMed Central articles (*Tahsin et al., 2016*). Our approach is distinguished from this in two ways. Firstly, we start from sequences rather than records, allowing us to examine the distribution of environmental contexts within a certain level of sequence similarity. Secondly, we associate to EnvO terms rather than to geographic coordinates. This makes seqenv more relevant to exploring the ecology of microbes, determining the distribution of OTUs across environment types, as opposed to tracking viral outbreaks which was a previous focus (*Tahsin et al., 2016*). The information that seqenv automatically generates can answer similar questions to those addressed previously (*Chaffron et al., 2010*), where co-occurrence of OTUs across sampling sites was examined and isolation sources were classified to EnvO terms through text matching. We provide this functionality in a single coherent software pipeline and promote user-friendliness.

To illustrate the usefulness of our pipeline, we apply it to two different datasets. The first is a previously published study of AOA derived from 45 soils (*Gubry-Rangin et al., 2011*). These soils present a range of pHs, enabling us to uncover how the spectrum of environments from which these organisms derive varies with changing pH. The second dataset is from sediment cores deriving from the Black Sea (*Coolen et al., 2013*). Here the 18S rRNA gene was sequenced by targeted-metagenomics, determining the eukaryotic plankton community structure over the last twelve thousand years. We can use seqenv to relate the environmental preferences of these organisms to changes in Black Sea geology, most notably the initial Mediterranean sea influx (IMI), hence, providing insight into the Black Sea environment prior to the IMI event.

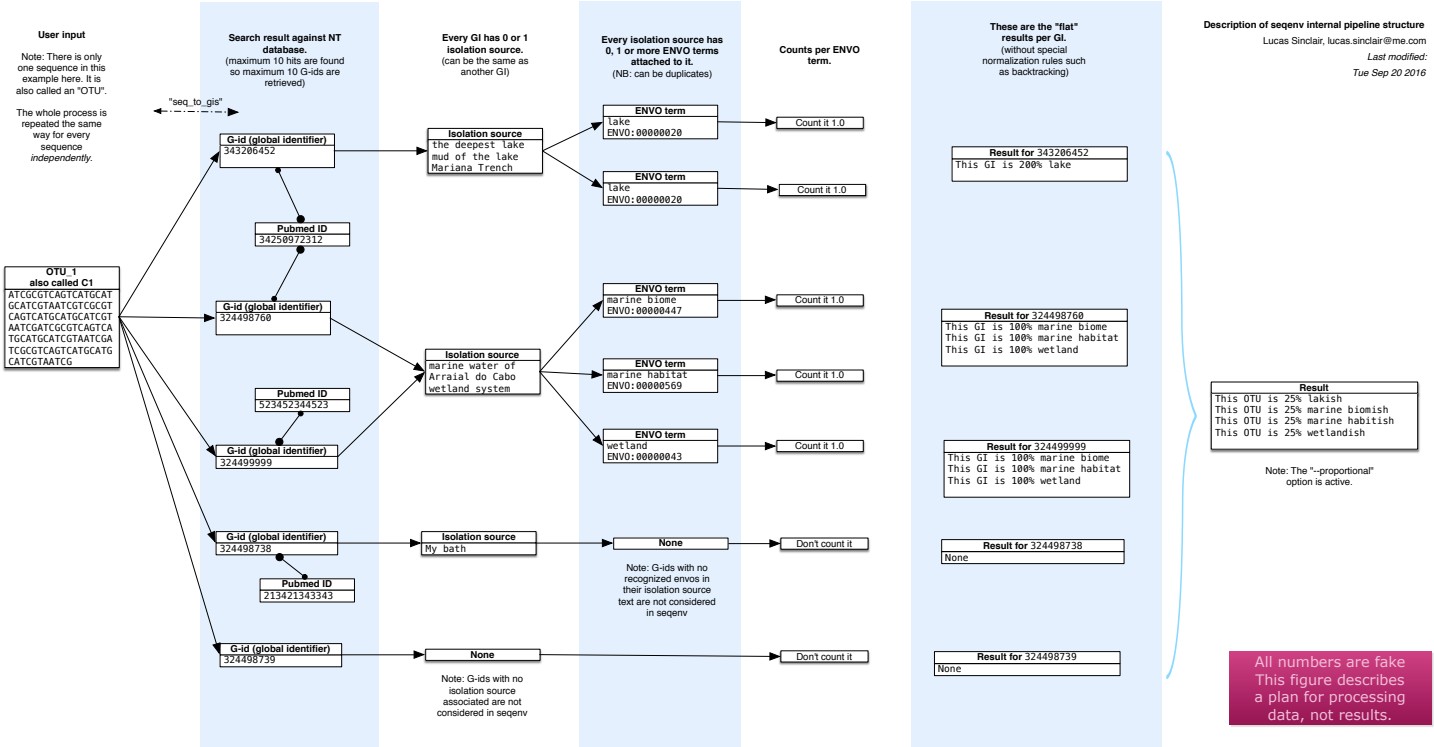

**Figure 1** **Schematic of the internal functioning of the seqenv pipeline.** This figure details how EnvO term frequencies are computed. The numbers provided are fictional as the schematic focuses on representing the internal functioning of the pipeline and does not illustrate a concrete case. As each inputted short DNA sequence is processed independently in all but the last stages of seqenv, only one input sequence is shown here.

## MATERIALS AND METHODS

The seqenv pipeline proceeds through the following steps, as illustrated diagrammatically in Fig. 1. The input is a user-supplied FASTA file containing thousands of DNA sequences and, optionally, a frequency file containing the frequency counts of the sequences across multiple samples. This file takes the form of a tab delimited text file containing the count matrix. In typical usage, the sequences would correspond to the consensus sequences of OTUs and the matrix would represent their frequencies across samples. After the following procedure, multiple outputs are generated:

1. The first step that seqenv executes is the parsing of the FASTA input file. All the sequence names are removed and replaced by a place-holder title following the sequence "C1", "C2", "C3", etc. In this fashion, problems caused by odd encodings or ambiguous characters are circumvented.

2. The second step consists of an optional filtering of the sequences to include only the most abundant i.e., highest total frequency across samples. As the computation time scales with the number of inputs, this filtering can greatly increase performance while leaving results statistically unaffected. The number of selected sequences is a customizable parameter and the default is to use all sequences. If no frequency matrix is provided this step is skipped.

3. Next, every remaining sequence is compared to a database of the user's choice. By default, the "nt" (nucleotide) database provided by NCBI is used and the BLAST algorithm is chosen to carry out the similarity search (*Altschul et al., 1990*). This step is the most costly computationally. It can, however, be parallelized by seqenv on multi-core systems by performing a simple and automatic input-chopping strategy.

4. Taking all the results from the sequence similarity search, the best hits are selected by filtering them according to the *e*-value of the comparison, the coverage of one sequence against the other, the identity between one sequence and the other, and a maximum number of targets for each input sequence. These parameters default to 0.0001, 0.97, 0.97, and 10 respectively.

5. For every search hit from every input sequence, the corresponding GenInfo Identifier (GI) of the homologous target within the database is recorded. This creates a table that links every input sequence to zero, one or more GI numbers.

6. Then, we collect the "isolation source" text entries associated to all of the GI numbers recorded in the previous step, provided the GI number was associated with such a field in NCBI's database, failing which it is discarded. No internet connection is required as all text entries are stored in an SQLite3 database and can be accessed locally by seqenv. This database links every GI number to its PubMed identifier, along with its isolation source text.

7. Using all the isolation source texts collected in the previous step and a text mining module, we proceed to identify all terms that contain some type of environmental information. Words such "glacier", "pelagic" or "forest" are extracted and connected to the controlled EnvO vocabulary. This consists of a hierarchically organized network of descriptive terms. In particular, the frequency of occurrence of each word is noted. Concretely, this is done offline by using a named entity recognition (NER) system. The NER algorithm is an optimized dictionary-based tagger, it searches for keywords associated with each ENVO term but also using a stop-list of problematic words (*Pafilis et al., 2015*). The ability of the NER engine to tag text with ENVO terms was evaluated in *Pafilis et al. (2015)* through comparison to a manually curated corpus this resulted in 87.8% precision and 77.0% recall, corresponding to an F1 score of 82.0%. The results were placed into an SQLite3 database that is automatically downloaded on the first run of seqenv.

8. With all the computed information, we are now able to describe each input sequence by a set of EnvO terms and their associated frequency forming a term-frequency vector. Across the whole dataset, this forms a sequence-term matrix. This matrix $\mathbf{S}$ has elements $s_{j,k}$ given the weight of the $k$th EnvO term associated with the $j$th sequence. These weights are calculated according to three different normalization strategies. The first is named "flat" and consists of using the raw occurrence counts. The second is termed "unique isolation" and will count every identical isolation source only once within the same input sequence, removing duplicated entries. The third is titled "unique pubmed unique isolation" and will uniquify the frequency counts based on the text entry of the isolation sources, as well as on the PubMed identifiers from which the GIs are obtained, removing all but one matching sequence in the event they pertain

to the same study. In all cases, the rows of the matrix are normalized to 1.0, such that $s_{j,k} = s'_{j,k}/\sum_l s'_{j,l}$, where we are denoting the raw counts by $s'_{j,k}$. The default normalization strategy is "flat".

9. If the user supplied a frequency matrix (c.f. second step), we are able to describe every one of the original biological samples by a set of EnvO terms and frequencies that are simply the sum of the term vectors over all sequences, weighted by the abundance of that sequence in the sample. Equivalently, the sample term matrix $\mathbf{N}$ elements $n_{i,k}$, is the matrix product of the frequency matrix $\mathbf{F}$ elements $f_{i,j}$ and the sequence-term matrix, i.e., $n'_{i,k} = \sum_j f_{i,j} s_{j,k}$. Normalizing by the total frequency in the sample, such that $n_{i,k} = n'_{i,k}/\sum_l n'_{i,l}$, we obtain sample term vectors such as, translated to English: "Sample Z is 25% brackish estuary, 25% river and 50% wetland".

10. Other options are available to the user to further modify and filter the results. The "backtracking" option, when activated, will propagate frequency counts up the acyclic directed graph described by the ontology for every EnvO term identified by the text mining module. The "restrict" option, when specified by passing a given EnvO identifier (e.g., ENVO:00010483), will force the output to contain only descendants from a single EnvO term. In effect, all other terms that are not reachable through the given node in the ontology graph are removed.

11. The first output that is produced is a table serialized in the format of a tab-delimited plain text file (TSV) representing the composition of each input sequence according to the EnvO terms associated to them, i.e., the matrix $\mathbf{S}$. The columns represent input sequence and rows represent the normalized weight of EnvO terms.

12. If the user provided a frequency matrix (as described in step 2), the program can produce a similar TSV table representing the composition of each biological sample according to the EnvO terms associated to them, i.e., the matrix $\mathbf{N}$. In this case, columns represent samples and rows represent EnvO terms. Each value corresponds to the normalized weight of the EnvO term in the corresponding sample.

13. For each sample, a visual representation of the hierarchy of the EnvO terms occurring in the isolation source of its imputed close relatives can be made. A PDF file is generated for each sequence and, if the user provided an abundance table, for each sample. In addition, every PDF has a corresponding DOT file which can be viewed and manipulated with the Graphviz software.

14. Other intermediary outputs are available as well, such as the output of the similarity search and a precise list of every EnvO term found in each input sequence.

The seqenv package is written in Python. The code follows a clean architecture, is commented and object-oriented. It is free and open-source carrying an MIT license. It is available on github here: https://github.com/xapple/seqenv. It can be installed on any computer with Python by simply typing: "pip install seqenv" in your shell.

## RESULTS

Earlier versions of the seqenv pipeline have already been used in a number of published studies including an analysis of the degree of recruitment of marine bacteria from freshwater

sources and the air (*Comte et al., 2014*), as well as a survey of bacterial diversity along a 2,600 km river continuum (*Savio et al., 2015*), and a study of hydrogenase genes in lake sediments (*Couto et al., 2015*).

Here, to further illustrate its utility, we will apply it to two published datasets and demonstrate that it provides additional insights into the processes that structure microbial communities not evident in the original analyses. These two examples comprise:

1. A survey of archaeal *amoA* gene data from 45 British soils, originating from a broad range of pH (min. 3.5, max. 8.7, median 6.2) (*Gubry-Rangin et al., 2011*). The sequences were generated by bidirectional 454 pyrosequencing of part of the *amoA* gene, reads were denoised with AmpliconNoise (*Quince et al., 2011*), overlapped and further error checked by removing those with stop codons when translated into amino acids. For this part of the analysis, we generated operational taxonomic units (OTUs) at 5% sequence divergence using average linkage hierarchical clustering. This will be higher resolution than species (*Pester et al., 2011*), corresponding to ecotypes with well defined environmental preferences. This procedure resulted in just 67 OTU sequences. All sequences were from archaeal ammonia oxidizers (AOAs) as described previously (*Gubry-Rangin et al., 2011*).

2. The Black Sea Paleome. This study included 454 pyrosequencing of 18S rRNA gene amplicons from 48 deep sediment samples collected from the Black Sea enabling the reconstruction of microbial eukaryote populations up to 11,400 years in the past. The V1–V3 region was sequenced as described previously (*Coolen et al., 2013*). Reads were denoised with AmpliconNoise and OTUs constructed at 3% sequence divergence using average linkage hierarchical clustering as species proxies (*Quince et al., 2011*). A total of 1,748 OTUs were obtained.

## Patterns of ammonia oxidizing archaea (AOA) habitat usage

In total 67 OTUs were observed across the 45 samples. These OTUs have been previously demonstrated as having well defined pH preferences (*Gubry-Rangin et al., 2011*). For each OTU, we calculated the mean of its pH range as the weighted averaged of the samples it was observed in, i.e.,:

$$\bar{Y}_s = \sum_{n=1}^{N} x_{n,s} Y_n,$$

where $\bar{Y}_s$ is the mean pH range for OTU $s$ and $x_{n,s}$ is the relative abundance of $s$ in sample $n$, which has pH $Y_n$. We ran `seqenv` on the 95% OTU centroid nucleotide sequences considering up to 100 matches with 95% overlap and 95% identity to the query. Once again, this procedure should return all sequences within approximate species boundaries. The default "flat" normalization was used. We restricted the analysis to all EnvO terms that inherit from the term "environmental material" which is identified by the number ENVO:00010483. Thereby, the redundancy across different terms in our analysis was reduced. In Fig. 2, we show the EnvO terms associated with two OTUs deriving from the extremes of the observed pH ranges (C46—3.5) and (C66—8.5). These OTUs had

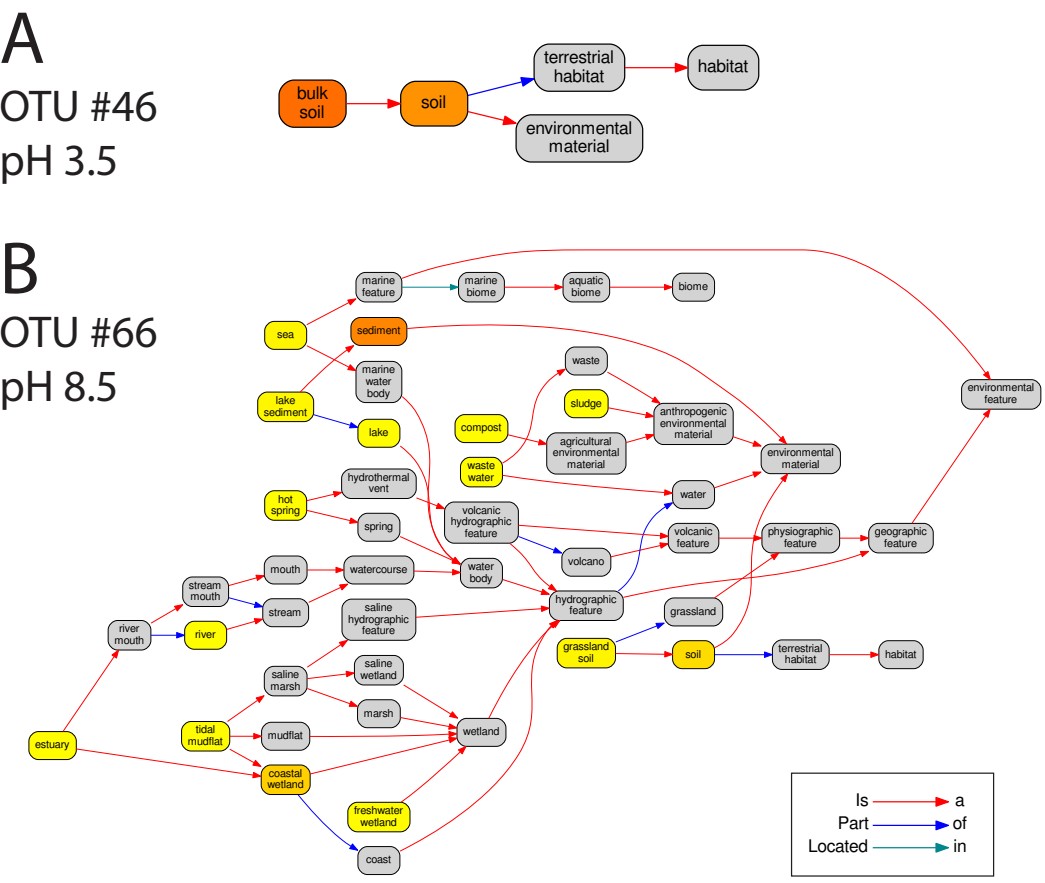

**Figure 2** **The EnvO terms associated with two AOA OTUs.** For each original inputted sequence, seqenv outputs a network representing the EnvO terms identified. Two examples of such hierarchical ontologies are shown. The two OTUs chosen had a mean pH of 3.5 and 8.5. The intensity of the node's background color reflects the frequency of that term within hits. Gray indicates the lowest frequency recorded and darker shades of yellow to orange indicate higher frequencies.

two and fifteen EnvO terms associated with them in total respectively. In all, we obtained EnvO terms for 66 OTUs. The 67th OTU did not match to any sequences carrying environmental information in the database. In Fig. 3A, the total number of terms found for each OTU as a function of its preferred pH range is plotted. A significant positive correlation between the diversity of habitats and the pH of the samples the organism was found (adjusted $R$-squared: 0.274, $p$-value: 3.85e−06). Another weaker but still significant positive association is observed between sample pH and total OTU diversity (adjusted $R$-squared: 0.131, $p$-value: 0.00922).

To determine which EnvO terms were most associated with the pH preference of the AOA OTUs, we performed a Random Forest regression (*Breiman, 2001*) of pH preference against the weighted EnvO terms. Random Forest uses an ensemble of decision trees constructed from artificial datasets generated by bootstrap aggregation or <bagging>, i.e., sampling with replacement across samples. This is combined with random selection of features. Since not all samples are used in each dataset, a robust estimate of model accuracy

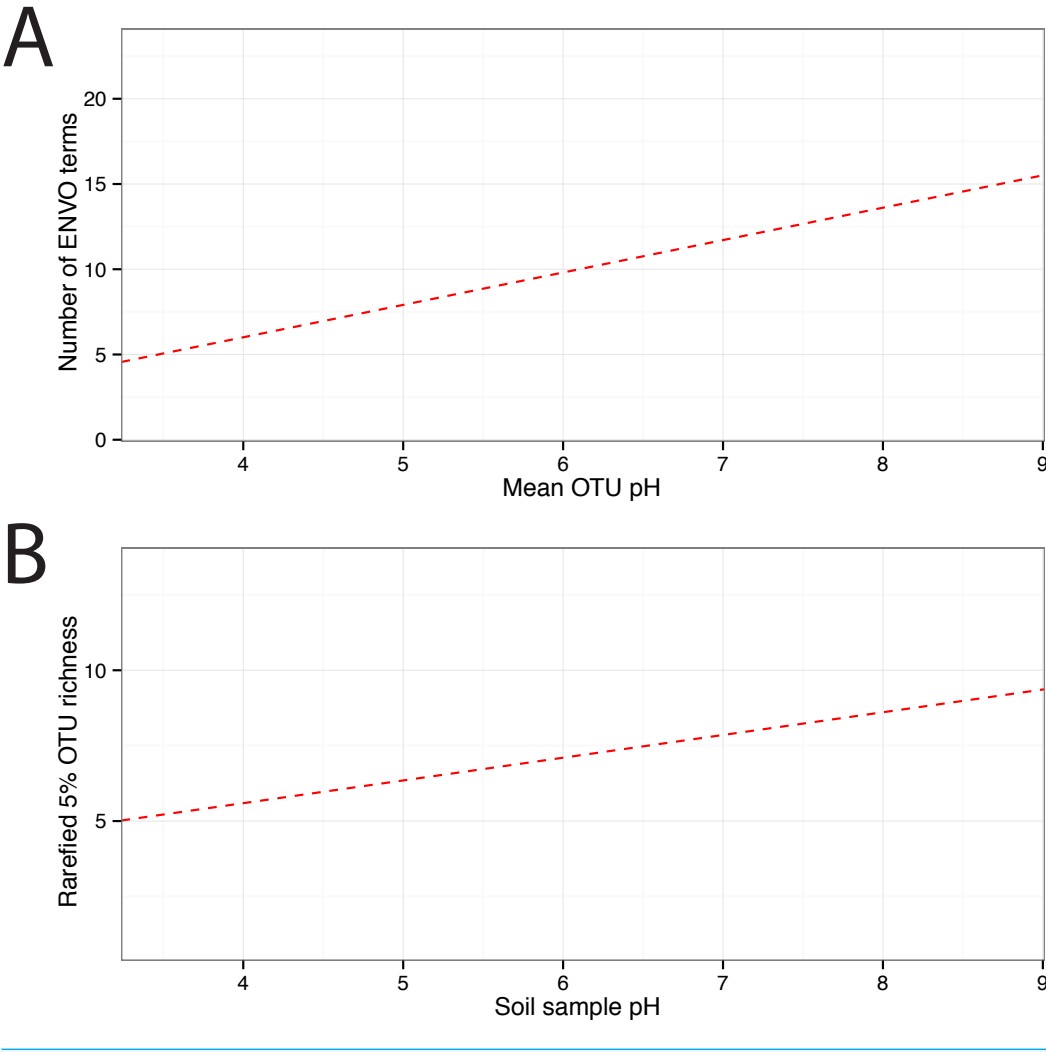

**Figure 3  EnvO terms and OTU richness against mean OTU pH.** (A) shows the total number of EnvO terms against OTU pH. The dashed red line indicates a linear regression of number of EnvO terms with OTU pH (adjusted *R*-squared: 0.2742, *p*-value: 3.85e−06). (B) shows the community OTU diversity against sample pH for the AOA dataset. OTU richness was calculated after rarefying to 1,000 reads. Linear regression of sample diversity against pH (adjusted *R*-squared: 0.1305, *p*-value: 0.009217).

is possible using the left out samples. Additionally, estimates of variable importance can be obtained by comparing accuracy of prediction with and without randomly permuting the variable of interest. This is measured by the statistic: percentage mean decrease of accuracy (*%IncMSE*). We fitted a Random Forest using the `randomForest` R package (*R Core Team, 2014*; *Liaw & Wiener, 2002*). The model explained 34.6% of the variation in pH preference. In Fig. 4, we visualize the weights of the top ten most important terms as determined by *%IncMSE* across OTUs ordered by their pH preference.

## Environmental stages of the Black Sea paleome

The 48 sediment samples form a series from a Black Sea core spanning the last 11.4 thousand years before present (kyBP). In *Coolen et al. (2013)*, they defined four "Environmental

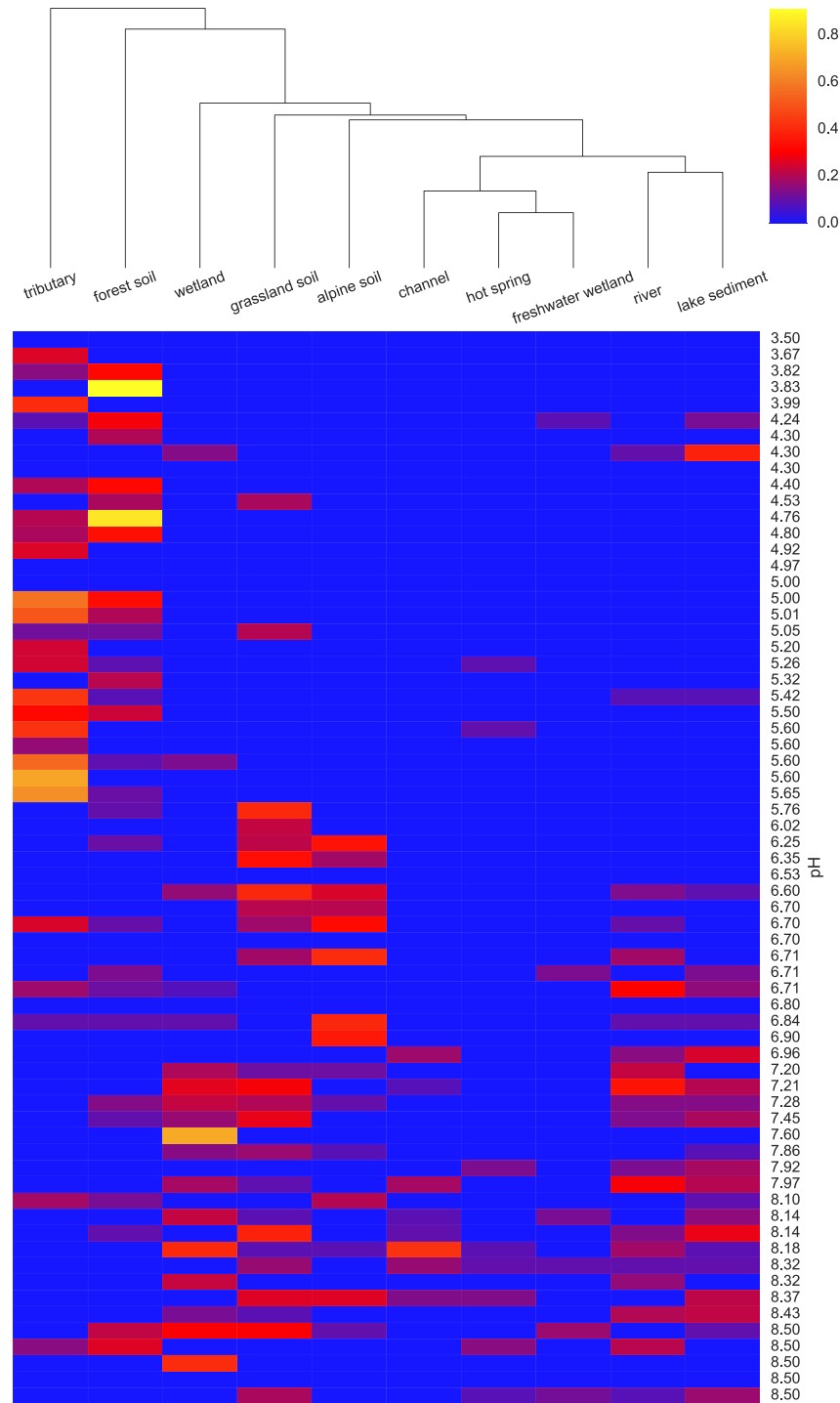

**Figure 4   Heatmap of top ten EnvO terms for determining OTU pH.** Random forests were used to perform a regression of pH against EnvO terms (Var. explained: 34.6%). The abundance of the top ten most important terms, as determined by percentage mean decrease of accuracy (%*IncMSE*) are shown across OTUs ordered by their pH preference.

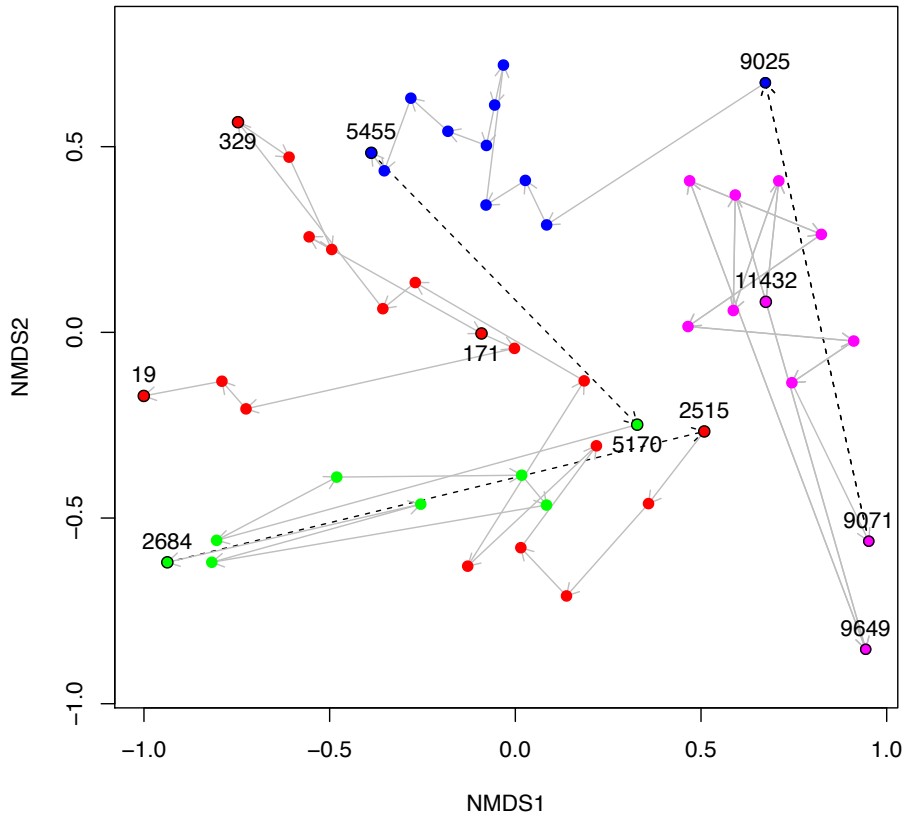

**Figure 5** **NMDS plot of Black Sea plankton 18S rRNA samples.** Non-metric multidimensional scaling (NMDS) of OTU relative abundances with Bray–Curtis distances were used to ordinate the 18S rRNA Black Sea plankton samples in two dimensions. The age of key samples are indicated together with the Environmental Stage: ES4 (magenta), ES3 (blue), ES2 (green) and ES1 (red). Arrows indicate the temporal succession of samples, and dotted arrows represent the transition between environmental stages.

Stages" (ES) in the geological evolution of the Black Sea that apply to this depth series on the basis of fossil evidence and isotope ratios:

- ES4: Lacustrine interval (∼11.4–9.0 kyBP). During this lacustrine phase the Black Sea was disconnected from the Mediterranean Sea due to low sea levels. This phase ends with the initial marine inflow (IMI) as rising sea levels, due to the end of the ice age 11,700 years ago, resulted in the connection of the Black Sea to the Mediterranean.
- ES3: A period of increasing salinity (∼9.0–5.2 kyBP) corresponding to the warm and moist mid-Holocene climatic optimum.
- ES2: Establishment of modern environmental conditions (∼5.2–2.5 kyBP) and further increasing salinity associated with the onset of the dry Subboreal.
- ES1: Freshening (∼2.5 kyBP to present) with onset of the cool and wet Subatlantic climate and recent anthropogenic perturbations.

In Fig. 5, we visualise the community structures of these samples, in terms of the 18S rRNA OTU proportions using a 2D non-metric multi-dimensional scaling (NMDS). This is very similar to Fig. 2A of *Coolen et al. (2013)*, except that the OTUs in our study were

constructed differently, but we include it here for the sake of completeness. The trajectory through time of the samples together with their Environmental Stages are shown. From this it is clear that there is a coherent change in structure during the geological history of the Black Sea and that the samples cluster according to ES.

We next ran seqenv on the 1,748 18S rRNA OTU centroid sequences taking into account up to 100 matches with 97% overlap and 97% identity to the query. As above, we restricted the analysis to those terms that inherit from the term ENVO:00010483 "environmental material" and used the "flat" normalization option. The normalized term vectors for each OTU were then combined with the relative OTU frequencies across the 48 sediment samples to obtain the weighted frequency of terms across samples, as described above. In total we observed 99 separate EnvO terms across the 48 samples. As above, we used a random forest classifier to predict these environmental stages from the EnvO terms associated with each sample. This classifier had an error rate of 12.5%. In Fig. 6 we show the relative frequency of the ten most important terms in this classifier across the samples, ordered by age and with the ES groups indicated.

## DISCUSSION

The two analyses presented above demonstrate the value of using seqenv to associate EnvO terms with both individual OTUs and whole samples. In the analysis of AOA OTUs, we demonstrated a significant association between the pH that an OTU is adapted to and the diversity of environments that it is found. These results indicate that, as their optimum pH increases, the AOA OTUs are present across a greater diversity of habitats. As in the original study, a statistically significant relationship between sample OTU richness and pH was observed. That is, as the pH of a sample increases, more species are observed. We propose that these two observations may be connected: the fact that more environments appear accessible to the OTUs as the pH increases may generate the diversification of species that is reflected in the increasing sample richness with pH. At higher pHs, we might expect both of these relationships to be reversed due to increased competition with bacterial ammonia oxidizers.

In the geological history of the Black Sea, one of the key questions is the nature of that environment prior to the initial Mediterranean sea influx (IMI). For example, was it a Brackish environment, or was it akin to a freshwater lake landscape? In our Black Sea dataset analysis, we can note a discrete change in the EnvO terms associated with the samples at this event when we transition from ES4 to ES3. Prior to this point, terms such as "freshwater lake" and "river" are frequent, afterwards the samples are dominated by organisms associated with "sea water", "ocean water" and "estuary". The microbial community prior to the IMI comprised organisms associated with freshwater habitats, important evidence that the IMI was associated with a substantial increase in salinity.

## CONCLUSION

The two studies described in this paper are not intended to be exhaustive, but present convincing vignettes of the usefulness of seqenv. We believe the methods presented here will prove to be an effective and extremely valuable tool to the community for distilling,

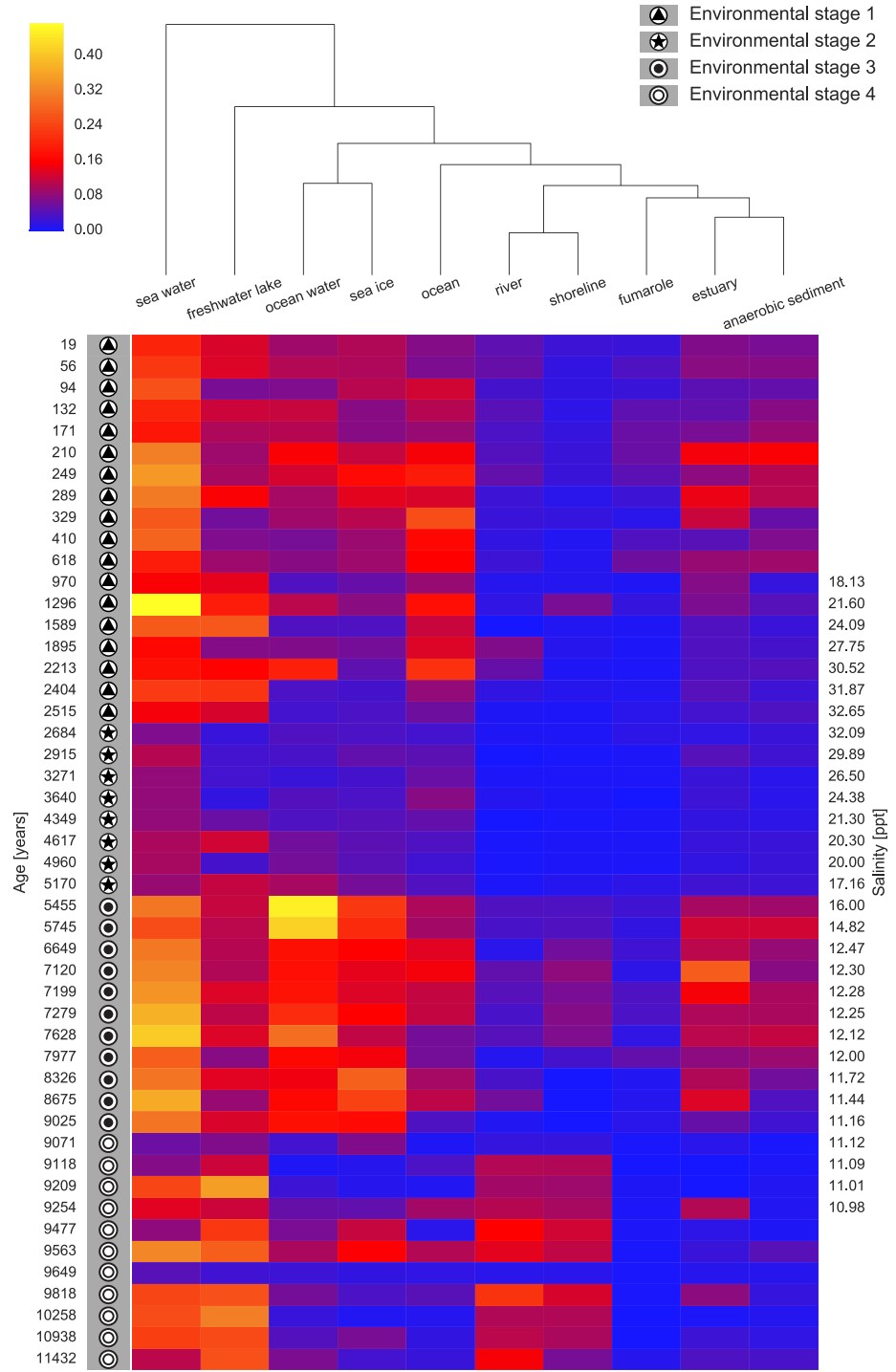

**Figure 6  Heatmap of top ten EnvO terms for determining ES.** Random forests were used to perform a classification of the Black Sea Environmental Stage (ES) against EnvO terms (Error rate: 12.5%). The abundance of the top ten most important terms, as determined by the percentage mean decrease of error rate (%IncMSE) are shown. Samples are labelled with ES and order by time before present, and salinity is given for the central part of the sediment core.

analyzing and adding context to DNA sequence data. There may be areas in which our methodology could be improved for example by weighting terms in OTUs or samples using more sophisticated approaches from information retrieval. In any case, we believe that, in the future, seqenv will contribute crucial insights and advances to the field of environmental metagenomics.

## Abbreviations

| | |
|---|---|
| **16S** | 16 Svedberg sedimentation mark (non-SI unit) |
| **ANOVA** | Analysis of variance |
| **AOA** | Ammonia oxidising archaea |
| **BLAST** | Basic local alignment search tool |
| **BP** | Base pair (of nucleotides) |
| **EnvO** | Environmental ontology |
| **GI** | GenInfo identifier |
| **HTS** | High-throughput (genetic) sequencing |
| **MSE** | Mean squared error |
| **NCBI** | National center for biotechnology information (in the US) |
| **NER** | Named entity recognition |
| **NMDS** | Non-parametric multidimensional scaling (ordination plot) |
| **OTU** | Operational Taxonomic Unit |
| **PCR** | Polymerase chain reaction |
| **rRNA** | Ribosomal ribonucleic acid |
| **TSV** | Tab separated values |

# ACKNOWLEDGEMENTS

Seqenv was originally conceived in a series of <hackathons> supported by the European Union's Earth System Science and Environmental Management COST Action. This project was titled ''Microbial ecology & the earth system: collaborating for insight and success with the new generation of sequencing tools'' and can be viewed at http://www.cost.eu/domains_actions/essem/Actions/ES1103. We would like to thank the LifeWatchGreece project (http://www.lifewatchgreece.eu/) for their generous support in the organization of these meetings.

## Funding

Lucas Sinclair and Alexander Eiler were funded by the Swedish Foundation for strategic research (ICA10-0015). Umer Zeeshan Ijaz was funded by NERC IRF (NE/L011956/1). Lars Juhl Jensen was funded by the Novo Nordisk Foundation (NNF14CC0001). Evangelos Pafilis was supported by the European Commission FP7-REGPOT project MARBIGEN (grant agreement #264089) and the LifeWatchGreece Research Infrastructure (384676-94/GSRT/NSRF C&E). Christopher Quince is funded through the MRC Cloud Infrastructure for Microbial Bioinformatics (CLIMB) project (MR/L015080/1) through

fellowship (MR/M50161X/1). Cecile Gubry was funded by the Environment Research Council Fellowship (NE/J019151/1). The funders had no role in study design, data collection and analysis, decision to publish, or preparation of the manuscript.

### Grant Disclosures

The following grant information was disclosed by the authors:
Swedish Foundation for strategic research: ICA10-0015.
NERC IRF: NE/L011956/1.
Novo Nordisk Foundation: NNF14CC0001.
European Commission FP7-REGPOT project MARBIGEN: #264089.
LifeWatchGreece Research Infrastructure: 384676-94/GSRT/NSRF C&E.
CLIMB project (MR/L015080/1): MR/M50161X/1.
Environment Research Council Fellowship: NE/J019151/1.

### Competing Interests

The authors declare there are no competing interests.

### Author Contributions

- Lucas Sinclair analyzed the data, wrote the paper, prepared figures and/or tables, reviewed drafts of the paper, wrote the software product seqenv in Python in its entirety.
- Umer Z. Ijaz analyzed the data, wrote the bash-based original version of seqenv until version 0.8.0, after which, LS restarted the implementation. Tested the software.
- Lars Juhl Jensen contributed reagents/materials/analysis tools, reviewed drafts of the paper, developed the NER software that seqenv relies on, as well as helped with using and installing it.
- Marco J.L. Coolen conceived and designed the experiments, reviewed drafts of the paper, supplied the Black Sea dataset.
- Cecile Gubry-Rangin conceived and designed the experiments, reviewed drafts of the paper, provided AOA data and expertise.
- Alica Chroňáková and Julia Schnetzer reviewed drafts of the paper, participated in the first hackathon.
- Anastasis Oulas reviewed drafts of the paper, helped test the software.
- Christina Pavloudi reviewed drafts of the paper, participated in the hackathons, helped test the software.
- Aaron Weimann reviewed drafts of the paper, participated in the second hackathon.
- Ali Ijaz participated in the second hackathon.
- Alexander Eiler reviewed drafts of the paper.
- Christopher Quince analyzed the data, wrote the paper, prepared figures and/or tables, reviewed drafts of the paper.
- Evangelos Pafilis analyzed the data, reviewed drafts of the paper.

### DNA Deposition

The following information was supplied regarding the deposition of DNA sequences:
   We used the ''NT'' database from NCBI and the associated GenBank records.

## Data Availability

https://github.com/xapple/seqenv.

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
