# Peer review of "Seqenv: linking sequences to environments through text mining"

_PeerJ, doi:10.7717/peerj.2690_

## Round 0.1 · original submission · Minor Revisions

· Academic Editor

Minor Revisions

The reviewers were unanimously positive about the quality of the article and requested relatively minor revisions. Therefore, I do not see the need to delay the manuscript by returning to the reviewers if you can respond to all their points. Please respond to each issue raised and detail changes made in your rebuttal letter. I will assess whether the new version has satisfactorily addressed their suggestions.

·

Basic reporting

This manuscript describes a novel approach to analyzing microbial metagenomes. The authors compare each read in the metagenome to the standard libraries, and then extract the environmental records associated with each of the sequences. They present those records as evidence of where the sequences likely came from.

The manuscript describes the pipeline that they have developed, the approaches that they use, and the summary statististics that are generated. In addition, the manuscript includes two use cases that demonstrate the utility of this approach.

Experimental design

There was a minor issue in downloading and installing the software - graphviz and the development version of graphviz needed to be installed before the pip install seqenv command would successfully complete.

The approach that is used is to compare the sequences to a standard database using BLAST. seqenv can handle parallelization of the BLAST search. This is not described but presumably it is via the number of threads option to the blast executable. However, BLAST is very slow and there are better search algorithms available these days (e.g. lastal, rapsearch2, diamond), and it would be useful to either accept one of the standard output formats that is (approximately) supported by those applications (e.g. tab-separated output) or to provide the search executable as an option to the seqenv program.

Validity of the findings

seqenv represents a useful tool to summarize the environmental data associated with metagenomes, and will be useful as an exploratory tool to analyze the data.

Additional comments

I have made several comments in the pdf

·

Basic reporting

• Some of the acronyms used in the Introduction should be defined for readers unfamiliar with the field (i.e. NCBI, BLAST, PCR, etc.)
• EnvO is used in line 74 but not defined until line 77
• In line 74 the authors reference citation #8, “A custom named entity recognition (NER) system based on [8] is then used to label the resulting text with terms from the EnvO ontology [3].” It would help the reader to be more clear in the text exactly what the NER system is based on rather than simply referencing the citation.
• In line 82 the authors state, “We can apply this environmental annotation scheme to any type of sequence, protein coding or ribosomal rRNA.” Statement would be stronger if personal pronoun ‘we’ is removed, “This environmental annotation scheme can be applied to any type of sequence, protein coding or ribosomal RNA.” Should say ribosomoal RNA or rRNA, not both.
• Since they’re both short paragraphs, the authors could combine the paragraphs starting on line 82 and line 86.
• On line 96, “Firstly, we start from sequences rather than records, allowing us examine the distribution of environmental contexts within a certain level of sequence similarity, secondly we associate to EnvO terms rather than geographic coordinates.” Should have a ‘to’ in between “…allowing us” and “examine the distribution…”
• I also recommend the authors separate the last sentence into two sentences “…certain level of sequence similarity. Secondly, we associate…”
• Why are there question marks in citation brackets for Random Forest in lines 237 and 244?
• Figures and Tables should be cited according to PeerJ guidelines.
• Figure 2 should be reformatted maybe as a panel? Panel A for the top figure about pH 3.5 and panel B for the bottom figure about pH 8.5. Same for Figure 3, the authors reference top and bottom panel in the text description but it will be visually clearer if they make it Figure 3A and Figure 3B.
• Figures 4 and 6 legends need to be reformatted so that they are under the figure and on the same page.
• Reformat in-text citations according to PeerJ guidelines.
• Per PeerJ guidelines please remove the funding agencies from the acknowledgements as they have a separate section for funding.
• It is recommended the authors read through the manuscript scanning for typos, properly defining abbreviations, and grammatical errors.

Experimental design

• Are there any plans for seqenv to take inputs other than FASTA? Such as FASTQ or compressed files (ex. .gz). While it is understandable to take in FASTA files as they are downloadable from NCBI, for ongoing studies that have sequenced data not yet submitted to GenBank, it would be nice to have opportunity to input a FASTQ file rather than have to convert to FASTA format.
• I would like to see some examples of the output generated from seqenv on the tool’s GitHub page, it may be there though I did not see example outputs, especially for the visualizations generated by Graphviz.
• The input for seqenv is a FASTA file and the authors reference its applications in metagenomics yet, the examples for the applications of the tool are only with 16S and 18S datasets. It is unclear if this tool can be used on true whole genome sequenced (WGS) metagenomics. If so, the authors should make this clearly stated and ideally, demonstrated on some shotgun WGS metagenomics data. Otherwise, the authors should reconsider some of their claims and statements throughout the manuscript.

Validity of the findings

The authors clearly demonstrate the applications of seqenv on the AOA and Black Sea datasets. I would like to see more evidence of their method applied to metagenomics data, especially as most of the tools used to analyze WGS metagenomes do not provide OTUs but tables listing organisms and taxa, as well as their relative abundance or number of reads.

Additional comments

In this paper Sinclair et al., present a novel method for linking sequences to environments. The authors should be commended for creating a tool that has major applications to the field of microbiology, addressing a concern of many scientists as microbiome and metagenomics studies become more global and widespread. The authors clearly demonstrate the efficacy of their approach on some datasets though, the application of the method to true metagenomics (i.e. whole genome sequencing) data needs to be demonstrated better.

Reviewer 3 ·

Basic reporting

The overall standard is very good. Some minor issues as below:

56: Delete comma after hitherto
64: fix wording - environments have these organisms been found in
66: a particular
208 et. seq.: missing refs

The software and documentation appear well put together.

Experimental design

The paper is well structured and the focus is clear. The technical standard of the work appears adequate and mostly of a very high standard, but it would be helpful if the authors were to include more detail of a couple of aspects as discussed in the general comments below.

Validity of the findings

The overall results appear valid and are of a standard appropriate for the journal. Some clarifications as described in the comments below would be helpful but are not sufficient to undermine the general view. The conclusions are measured and well described, and the approach is of broad interest.

Additional comments

The authors present a very interesting new pipeline for the annotation of genomic sequences extracted from diverse metagenomic samples by querying the metadata associated with the resulting species level BLAST hits. The methodology appears sound and the manuscript is in general well written, with a very clear positioning of the approach relative to recent alternatives involving direct geolocation.
The steps are laid out quite adequately in the methods section, with some good visuals supporting the explanation. Optional filtering to reduce execution time, the computational load perhaps being most troublesome in the BLAST based search step. It is not clear from the work presented whether the use of different BLAST parameters or much faster, lower sensitivity alternatives would have a significant effect on the resulting term calling. It would be a useful extension to this paper to explore this, as it is possible that the effect on accuracy of annotation may be modest relative to the performance gain. USearch is one such possibility.
The software is well organised and readily available from the github site with detailed instructions. Installation for linux systems is straightforward and windows users can probably use a guest environment successfully, though I did not try this.
It was not possible to assess the effectiveness of the NER work undertaken, though evidently the results are far from poor. I note the reliance on the earlier 2015 study which in its conclusions suggests just this application, but it would be nice to have some clearer idea of this part of the work.
The case studies are well presented but the biological context is outside my expertise.
The discussion of the normalisation alternatives is relatively light. At present, only simple approaches are employed, and there appears not to have been much effort to link this work to established ideas from information retrieval and related fields. While this work is a firm foundation, there may well be fruitful exploration possible in that direction.

---

## Round 0.2 · accepted · Accept

· Academic Editor

Accept

I am satisfied that all the reviewers points have been addressed and the manuscript is suitable for publication. I have a few small points:

1) On the marked PDF all the citations appeared to have been replaced by a "?" sign so I couldn't check them but I will assume they are all OK.

2) I will refer your question about adding abbreviations to the PeerJ staff. I am in favor of adding the list you provided in your response.

3) Typo on line 277